# The Impact of an Acrobatics-Based Curriculum on Motor Fitness in Adolescents

**DOI:** 10.3390/life15050778

**Published:** 2025-05-13

**Authors:** Saša Veličković, Edvard Kolar, Miloš Paunović, Petar Veličković, Saša Pantelić, Saša Bubanj, Igor Ilić, Bojan Bjelica, Tomislav Gašić, Adem Preljević, Ana-Maria Vulpe, Bogdan Constantin Rață, Cristina-Elena Stoica, Nicolae-Lucian Voinea, Tatiana Dobrescu

**Affiliations:** 1Faculty of Sport and Physical Education, University of Niš, 18000 Niš, Serbia; v.sale70@gmail.com (S.V.); zuxxx123@gmail.com (M.P.); gimnastika1997@gmail.com (P.V.); spantelic2002@yahoo.com (S.P.); 2Science and Research Centre Koper, 6600 Koper, Slovenia; edvard.kolar@zrs-kp.si; 3Faculty of Sport and Physical Education, University of Priština-Kosovska Mitrovica, 38218 Leposavić, Serbia; ilic.igor.dif@gmail.com; 4Faculty of Physical Education and Sports, University of East Sarajevo, 71126 Lukavica, Bosnia and Herzegovina; vipbjelica@gmail.com; 5High School Center Prijedor, 79000 Prijedor, Bosnia and Herzegovina; gasictomislav@yahoo.com; 6Faculty of Sport and Physical Education, State University of Novi Pazar, 36300 Novi Pazar, Serbia; apreljevic@np.ac.rs; 7Faculty of Movement, Sports, and Health Sciences, “Vasile Alecsandri” University of Bacău, 600115 Bacău, Romania; zaharia.ana@ub.ro (A.-M.V.); rata.bogdan@ub.ro (B.C.R.); lucian.voinea@ub.ro (N.-L.V.); tatiana.dobrescu@ub.ro (T.D.)

**Keywords:** physical education program, motor skill development, exercise intervention, physical activity, training adaptations, skill acquisition

## Abstract

(1) Background: This study aimed to examine whether an experimental acrobatics curriculum, conducted three times a week, could lead to greater improvements in specific components of motor fitness—coordination, balance, agility, and speed—compared to the standard physical education program; (2) Methods: The research was conducted over a 16-week period and included 50 seventh-grade students, divided into an experimental group (EG, *n* = 25) and a control group (CG, *n* = 25). The experimental group participated in a program of acrobatics and skipping twice a week, while the control group followed the standard physical education curriculum. Motor skills tests were administered before and after the intervention using standardized methods; (3) Results: Results from the ANCOVA analysis showed significant improvements in flexibility, coordination, balance, and agility in the experimental group, with large effect sizes, confirming the effectiveness of the acrobatics and skipping program. However, the impact on speed was variable, indicating the need for specific exercises to improve this ability; (4) Conclusions: The findings are consistent with previous research, highlighting the superiority of specialized acrobatic exercises in enhancing overall motor performance in adolescents. Further research is needed to optimize acrobatics and skipping programs for maximum benefits in the development of motor skills and physical education.

## 1. Introduction

Adolescence represents a critical period of intense physical and psychological development, during which significant changes occur in motor abilities such as strength, speed, coordination, and balance. These factors play a crucial role in promoting long-term health and well-being. Physical activity (PA) during adolescence has a profound influence on various anthropological parameters, including psychological (enhanced mental health and stress reduction), social (increased social interaction and teamwork), motor (development of coordination, strength, speed, and balance), and health-related aspects (prevention of chronic diseases and obesity) [1]. Studies consistently highlight the positive effects of diverse physical activities on adolescent health and fitness, emphasizing the importance of structured exercise programs in school curricula that significantly enhance overall motor competence in children and adolescents [2,3,4].

Gymnastics, as an integral part of physical education (PE), facilitates the development of fundamental motor skills and promotes a healthy lifestyle. It focuses on strength, flexibility, coordination, and balance—key components of overall physical development in youth [5].

Within the school PE framework, gymnastics provides opportunities to enhance motor abilities through diverse activities tailored to individual needs and capabilities. Studies indicate that educational gymnastics programs significantly improve motor skills and self-concept in children [6].

In this context, acrobatics-based training programs stand out as particularly effective for enhancing not only physical capacities such as strength, coordination, agility, and balance but also for improving self-perception and body confidence in children [6]. In addition to neuromuscular benefits, acrobatics programs can significantly improve muscular endurance and aerobic capacity through sustained, high-intensity routines that challenge cardiovascular function and build core and limb strength [7]. These benefits align with overall physical fitness goals and are particularly valuable for young people who may not engage in traditional endurance sports.

However, the level of PA among adolescents varies, and many do not meet the recommended guidelines, which can negatively impact their physical and mental development. The growing prevalence of a sedentary lifestyle among youth [8] underscores the urgent need for more proactive school programs promoting PA. Given its importance, PA plays a pivotal role in preserving adolescent health by preventing chronic diseases, enhancing cardiorespiratory fitness, and supporting healthy body composition [9].

Research indicates that adolescent girls tend to be less physically active than boys, which can contribute to disparities in health outcomes between sexes [10]. Early engagement in regular and diverse physical activities has been shown to establish a foundation for a more active and healthier lifestyle in later years [11]. Innovative approaches, such as physically active learning models, have also been effective in counteracting declining PA levels and improving cardiorespiratory fitness in adolescents [12].

School-based PE programs offer an ideal environment for promoting physical fitness among adolescents, as they provide access to structured PA regardless of academic performance or socioeconomic background [13]. Such programs are not only essential for motor development but also serve as a platform to improve social skills, emotional resilience, and mental capacity by fostering inclusion, peer interaction, goal setting, and discipline [14]. Additionally, school-based PE can serve as a preventive strategy for stress, anxiety, and other mental health concerns, which are increasingly prevalent in adolescent populations [15]. Furthermore, these programs ensure that all students, including those who are overweight or obese, can participate in organized physical exercise without experiencing discomfort or exclusion [9]. Given the increasing need for effective PE interventions, it is essential to implement strategies aimed at increasing adolescent PA levels.

The significance of this study lies in the necessity of providing empirical evidence to support the integration of acrobatics-based programs into the standard school curriculum. Such integration has the potential to enhance adolescents’ motor fitness, promote health, and improve body composition [16]. Moreover, as a dynamic and engaging form of exercise, acrobatics contributes not only to aerobic capacity when implemented in high-repetition, circuit-based formats [7], but also significantly boosts coordination and muscular strength [6], as mentioned above. This dual impact makes acrobatics an efficient and inclusive choice for schools aiming to improve students’ overall fitness within limited timeframes. However, there is a lack of recent meta-analyses and systematic reviews that could provide theoretical contrast or corroboration—particularly regarding school-based acrobatics programs.

It is important to note that a small portion of these result—primarily preliminary findings—were previously published in 2022 [17], focusing on select indicators related to body composition and flexibility. Previously published manuscript focused specifically on the effects of an acrobatics program on body composition and flexibility, while the current manuscript offers a broader investigation into motor fitness, including coordination, balance, agility, and speed, thereby expanding the scope and application of acrobatics-based interventions. Hence, the current work extends previous findings significantly, offering a more comprehensive dataset and deeper analysis. The broader scope and enriched statistical evaluation aim to contribute new insights to the existing literature and support the growing emphasis on acrobatics as a valuable form of physical activity in educational and developmental contexts. Readers are encouraged to refer to the earlier publication for context, while recognizing that the majority of the results and interpretations discussed herein are novel.

This study aimed to examine whether an experimental acrobatics curriculum, conducted two times per week, can lead to greater improvements in specific components of motor fitness—gross motor coordination (excluding balance, hereinafter referred to as coordination), balance, agility, and speed—compared to the standard PE program. The fundamental assumption was that the experimental acrobatics curriculum will lead to significantly greater improvements in flexibility, agility, and coordination compared to a standard PE curriculum prescribed by the Ministry of Education, Science, and Technological Development of the Republic of Serbia. Moreover, it was expected that balance would improve moderately, while the effects on speed would be variable, depending on the specific motor tasks tested. This hypothesis aligns with previous studies that emphasize the long-term impact of early PA on adolescent health and well-being [18].

The results of this research will provide valuable insights into the potential benefits of incorporating acrobatics into school PE programs and contribute to a better understanding of its role in improving the motor abilities of adolescents. The findings are expected to support the integration of comprehensive acrobatics programs into the standard school curriculum, enabling students to acquire skills and knowledge beneficial to their motor abilities.

## 2. Materials and Methods

### 2.1. Participant Sample

The study included 50 male seventh-grade students from the elementary school Bubanjski Heroji in Niš, Serbia, with an average age of 14 years ± 6 months. The participants were divided into two groups:Experimental group (EG, n = 25): Participated in an acrobatics program twice a week for 45 min per session, conducted by one PE teacher across 32 sessions.Control group (CG, n = 25): Followed the standard PE curriculum, which was implemented by a different PE teacher.

Participation in the study was voluntary, with informed consent obtained from both parents and the school.

#### 2.1.1. Inclusion Criteria

Participants were eligible for the study if they:Were males, 14 ± 0.5 years old,Had no chronic illnesses,Had no history of serious injuries.

#### 2.1.2. Exclusion Criteria

Participants were excluded from the study if they:Had respiratory or cardiovascular diseases,Had developmental disorders,Were in the process of recovering from injuries or illnesses,Had participated in organized physical activities within the six months prior to the study.Were taking any medication.

Only male participants were included in this study due to the differing physical development profiles and training needs of adolescent males and females, requiring adaptations in acrobatic training content and intensity. As the intervention was conducted during regular PE classes, it was not feasible for the PE teachers to simultaneously implement separate, tailored acrobatic programs for both sexes. The inclusion and exclusion criteria were assessed using a specially designed questionnaire.

#### 2.1.3. Randomization

Participants were randomly assigned to either the experimental (EG) or control group (CG) using a computer-generated randomization process to ensure equal distribution between the groups (Table 1).

### 2.2. Study Design

Motor abilities were assessed at the beginning and at the end of the experiment, which lasted for a total of 32 sessions (four months). EG participated in an acrobatics program (floor exercises and vault) twice a week for 45 min per session, totaling 32 classes, while CG followed the standard PE program (handball and volleyball) with the same frequency and duration.

To be included in the data analysis, participants were required to attend at least 80% of the sessions, equating to a minimum of 26 classes.

Before the experimental intervention, baseline tests were conducted to assess flexibility, coordination, balance, agility, and speed. Testing was performed over three days, with the same evaluators consistently measuring the same set of tests for all participants. After the experimental program, final testing was conducted under similar conditions.

Each session for both EG and CG was structured into four phases: introductory, preparatory, main, and concluding segments.

Before the study commenced, the experimental protocol and potential risks were explained to all participants, who provided informed consent. Additionally, parents provided written consent for their children’s participation. The study was approved by the University of Niš on 28 June 2019 (approval number 8/18-01-005/19-041) and conducted in accordance with the Declaration of Helsinki and ethical guidelines for research involving human subjects [19]. Participants were free to withdraw from the study at any time, either at their own discretion or at the request of their parents.

To maintain consistency, participants were asked not to alter their daily diet or lifestyle during the study period.

### 2.3. Variable Sample

Standardized methods and tests were used to evaluate flexibility, coordination, balance, speed, and agility. The instruments were carefully selected and adapted for this study based on previous research and expert recommendations.

The following tests were used to assess participants’ motor abilities, as described by Madić and associates [20], unless otherwise specified:

#### 2.3.1. Flexibility

Leg Raise from Supine Position (LRSP) Instruments:

Gym mat and a wooden board (300 × 150 cm) with a scale ranging from 0° to 180° in increments of 5°. The scale is positioned so that the x-axis is 10 cm above the bottom edge of the board, while the y-axis divides the board into two equal halves. Task: The participant lies on their back with their right side aligned with the board. They adjust their position so that their thighs are at a 90° angle. Arms are relaxed beside the thighs, legs together and fully extended. The right leg is slowly raised along the board to the maximum angle and held for a few moments. The test is repeated three times with breaks for measurement. Scoring: The result is the angle of the extended leg relative to the horizontal plane, expressed in degrees. All three results are recorded. The leg must remain fully extended, and the body must stay in contact with the mat; otherwise, the attempt is repeated. Note: Although the scale resolution is 5°, which is common in school settings for practical and observational ease, this level of granularity may limit the sensitivity in detecting subtle but meaningful flexibility changes.

Leg Abduction from Supine Position (LASP) Instruments: Gym mat and a wooden board (300 × 150 cm) with a scale ranging from 0° to 180° in 5° increments. The scale is positioned with the *x*-axis 10 cm above the board’s bottom edge, while the *y*-axis divides the board into two equal halves. Task: The participant lies barefoot on their back with their legs extended vertically and supported against a wall. The examiner ensures that the participant’s body center aligns with the goniometer’s axis of rotation. Upon the signal, the participant maximally spreads their extended legs apart. The knees must remain straight. Two correct attempts are performed with a 10 s rest between repetitions. Scoring: The result is the angle formed by the fully extended legs, expressed in degrees. The legs must remain fully extended, and the body must maintain contact with the mat; otherwise, the attempt is repeated. Similarly to the LRSP, the 5° increment scale may slightly limit the precision of the flexibility assessment, especially when capturing small improvements.

Tests that utilize goniometers or a drawn angular scale generally demonstrate high reliability and validity across most age groups, with reliability coefficients exceeding 0.80 [21,22].

Shoulder Rotation with Stick (SRS) Instruments:

A round stick (2.5 cm diameter, 165 cm length) with a plastic handle (15 cm) and a scale starting from 0 cm. Task: The participant stands holding the stick with their left hand gripping the plastic handle and the right hand placed immediately next to it. The stick is lifted above the head while separating the hands, keeping the left hand fixed. The goal is to rotate the stick with the smallest possible distance between the hands. The task is performed three times consecutively without breaks. Scoring: The distance between the inner edges of the hands, recorded in centimeters. All three results are recorded. Note: The arms must remain fully extended, and no momentum should be used; otherwise, the attempt is repeated. (Cronbach α > 90) [23].

#### 2.3.2. Coordination

Coordination was operationalized as the ability to efficiently perform complex, rhythmical, and multisegmental movements, distinct from balance, which was assessed separately.

20 Lunges with Stick Pass-Through (20LSPT) Instruments:

Wooden stick (30 cm length, 3 cm diameter), stopwatch (1/100 s). Task: The participant stands behind a marked line, holding the stick in their left hand. Upon signal, they step forward with the right foot, pass the stick under the leg, catch it with the right hand, and return to the starting position. They then repeat the movement with the left foot. This is performed 20 times. Scoring: The time taken to complete 20 correct repetitions is measured in seconds. Note: Incorrect movements do not count. The examiner counts aloud and warns the participant of errors.

Backward Obstacle Course (BOC) Equipment:

Swedish vault, electronic timing system with photocells (1/100 s). Space: Flat, smooth surface (12 × 2 m). A line is drawn 1 m from the start, and another line is drawn parallel at 10 m. At 3 m, a Swedish vault section is placed horizontally. At 5 m, the first vault frame is positioned. Task: The participant moves backward on all fours between the lines. The first obstacle is climbed over, the second is crawled under, without turning the head. The test follows a trial run. Scoring: Time is recorded from start to finish. If the participant knocks over an obstacle, they continue while the examiner resets it. Each participant has three attempts, and the median time is recorded. Note: If the second obstacle is knocked over before the feet pass, the participant must reset it and pass through again without stopping the stopwatch. The examiner checks obstacle placement.

#### 2.3.3. Balance

Y-Balance Test for Upper Body (YBTUB) Instruments:

Balance was assessed as a distinct dimension of postural control, evaluated in both upper and lower extremities using Y-Balance protocols.

Instruments: Balance Test Kit, consisting of a platform with three PVC tubes marked every 0.5 cm. Required tools include a measuring tape, paper, and pencil.

Task: The participant, in a supine position with legs spread apart and arms at hip-width, pushes the indicator in three directions (medial, inferolateral, superolateral). Each attempt is repeated three times per hand. Incorrect attempts are repeated.

Scoring: The maximum reach in all three directions is recorded and normalized according to the length of the limb. The formula for composite reach length is:Composite reach length = (Sum of the three reach lengths (cm))/(Sum of the three limb lengths (cm)) × 100.

Note: Participants perform the test barefoot. Measurements are precise to 0.5 cm. Incorrect attempts occur if the participant fails to maintain a unilateral position, loses contact with the indicator, uses the indicator for support, fails to return the hand under control, or lifts the foot.

The study showed that the YBT-UQ has high inter-rater reliability (ICC = 1.00) and test–retest reliability (ICC = 0.80–0.99) for different reach directions [24].

Y-Balance Test for Lower Body (YBTLB)

Instruments: Balance Test Kit with a platform and three PVC tubes marked every 0.5 cm.

Task: The participant stands on one leg, reaching as far as possible with the other leg in three directions: forward, backward-inner, and backward-outer. The order of testing is as follows: Right front reach, left front reach, right back-inner reach, left back-inner reach, right back-outer reach, left back-outer reach.

Scoring: The maximum reach is recorded and normalized according to the limb length. The formula for composite reach length is the same as in the previous case.

Note: The test is part of the 2015 NHL Combine protocol, based on the Star Excursion Balance Test. Participants perform the test barefoot after standardized instructions. Studies have shown that the Y-Balance Test (Lower Quarter) has high test–retest reliability, with Intraclass Correlation Coefficients (ICC) often exceeding 0.90, indicating high consistency between measurements. This test is reliable for assessing dynamic balance and can be used to identify asymmetries in lower limb function [25].

#### 2.3.4. Speed

Hand Taping (HT)

Equipment: Taping board (96 cm length, 12 cm width, 1 cm height), two round boards (20 cm diameter, 1 cm thickness), table (60 cm height), chair (40 cm height), and masking tape. Location: A room with a flat surface (2 × 2 m). The board is attached to the table parallel to the edge. The participant’s chair is on one side, and the examiner’s chair is on the other side of the table.

Task: The participant sits with their left hand placed in the center of the board and their right hand on the left board. At the “go” signal, the participant taps the boards alternately with their right hand as quickly as possible for 25 cycles. The task is performed after a practice attempt and stops when the examiner says “stop.”

Scoring: Time is measured in tenths of a second for 25 cycles of double tapping. If the participant fails to touch a board in a cycle, that cycle is not counted. The shortest time from three attempts is recorded. A study showed high test–retest reliability with intraclass correlation coefficients (ICC) ranging from 0.58 to 0.93, depending on the sample and specific test conditions [26].

20 m Sprint from a Standing Start (SSS20)

Equipment: Electronic timing system with photoelectric cells (Witty timer). Location: A flat, firm surface in a gym with minimum dimensions of 25 × 3 m. The start and finish lines are parallel and 1.5 m long. Mats are placed 5–6 m behind the finish line for stopping.

Task: The participant starts from a standing position and sprints 20 m, finishing when their chest crosses the finish line.

Scoring: Time is measured in hundredths of a second (1/100 s) from the start to when the chest crosses the finish line.

Note: The surface must not be slippery, and there should be no obstacles 5 m behind the finish line. An incorrect start is repeated. Studies have shown that the 20m Sprint Test is reliable and valid when using an electronic timing system. Research shows high test–retest reliability (ICC = 0.83–0.90) for 10 m and 20 m sprints when using dual photoelectric cells, with a coefficient of variation between 1.5% and 1.9%. This indicates a high level of consistency and accuracy in measuring short-distance sprint performance [27].

#### 2.3.5. Agility

Cone Drill 10x5 (CD10X5)

Description: Running and turning at maximum speed. Equipment: Non-slip surface, whistle, Witty photoelectric cells, meter, chalk or tape, cones.

Instructions for Participants: “Stand behind the start line with one foot behind the line. On the start signal, run as fast as possible to the second line and back, crossing both lines with both feet. Repeat this five times. On the fifth cycle, run at maximum speed to the end.”

Instructions for Measurement: Mark two parallel lines 5 m apart. The lines should be 1.2 m long and marked with cones. Ensure participants cross the lines with both feet. Time is recorded for ten 5 m sprints with accuracy to 1/100th of a second.

Scoring: The time for ten 5 m sprints is the result.

Studies have confirmed the reliability and validity of the 10 × 5 Cone Drill, which is part of the Eurofit test battery, indicating its broad application and acceptance in sports science.

### 2.4. Experimental Program

The experimental group followed an acrobatics program (floor and vault), while the control group continued with the regular curriculum (handball and volleyball).

Lesson Structure:

Warm-up (5 min): Identical for both groups, including walking, jogging, jumping, and basic ball games to prepare the body for exercise.

Preparation (15 min): Identical for both groups, consisting of stretching, strengthening, and mobility exercises to prepare muscles, tendons, and joints.

Main Part (20 min): Experimental group–Floor and vault exercises, performed in groups of three under instructor supervision, aiming at acrobatic skills, fitness development, teamwork, and social values. Control group–Standard handball and volleyball lessons according to the curriculum of “Bubanjskih Heroji” School in Niš.

Cool-down (5 min): Identical for both groups, featuring low-intensity exercises, relaxation, and recovery activities.

A summary table of lessons per group is included in Appendix A (Table A1).

### 2.5. Statistical Analysis

The analysis included basic descriptive statistics for the initial and final measurements in both the experimental and control groups. Inferential statistics comprised independent and paired *t*-tests to determine differences between and within the groups at initial and final measurements. Effect size (ES) was calculated to evaluate the magnitude of these changes. Additionally, analysis of covariance (ANCOVA) was used to assess the effects of the experimental treatment, adjusting for baseline scores. Effect sizes for ANCOVA were interpreted using Partial Eta Squared (η^2^), with thresholds defined as small (η^2^ = 0.01), moderate (η^2^ = 0.06), and large (η^2^ ≥ 0.14), as suggested by Richardson [28].

Cohen’s effect size was also used to interpret the magnitude of differences between pre- and post-test results within each group, following the classification by McGrath & Meier [29]: trivial (*r* ≤ 0.10), small (0.11–0.25), moderate (0.25–0.36), and large (>0.36). Statistical significance was set at *p* < 0.05. All analyses were performed using SPSS version 22.

A post hoc power analysis was conducted using the pwr.f2.test() function from the pwr package in R statistical software (version 4.4.0), assuming an alpha level of 0.05, a large effect size (f = 0.40, corresponding to η^2^ > 0.14), and two groups. The achieved power was calculated to be 0.87, exceeding the conventional threshold of 0.80, thus indicating adequate statistical power to detect large effects in the ANCOVA design.

## 3. Results

Table 2 presents the descriptive statistics and t-test results for the EG and CG in both the initial (pre-test) and final (post-test) measurements. The findings indicate notable improvements in several motor abilities in the EG, suggesting that the applied intervention was effective.

The results presented in Figure 1 indicate that the EG demonstrated significantly greater improvements across most measured variables compared to the CG, highlighting the effectiveness of the implemented intervention.

Table 3 offers a comprehensive breakdown of the statistical results, supporting the findings with precise values and tests for significance—it represents the results of the univariate analysis of covariance (ANCOVA), which examines the differences in motor abilities between the EG and CG after adjusting for initial measurements. The F-test values and significance levels (*p*-values) indicate whether the observed differences are statistically significant, while the Partial Eta Squared (η^2^) values provide insight into the effect size.

To ensure the validity of the ANCOVA results, key statistical assumptions were assessed and met. Specifically, normality of residuals was verified using the Shapiro–Wilk test, which showed no significant deviations (*p* > 0.05 for all outcome variables). Homogeneity of variances was confirmed through Levene’s test (*p* > 0.05), and the assumption of homogeneity of regression slopes was satisfied, as no significant interaction was found between the covariate (pre-test scores) and the group (*p* > 0.05). Additionally, considering the number of dependent variables analyzed, a Bonferroni correction was applied to adjust the significance threshold (α = 0.005 for 10 comparisons), and the most relevant outcomes remained statistically significant, further supporting the validity of the intervention’s effects.

To enhance interpretative precision, 95% confidence intervals were calculated for the differences between EG and CG in the post-test phase: LRSP (95% CI: 5.95 to 17.13), LASP (95% CI: 17.91 to 36.75), SRS (95% CI: −16.98 to −5.36), 20LSPT (95% CI: −13.81 to −7.17), BOC (95% CI: −1.88 to 6.66), YBTUB (95% CI: −4.63 to 6.55), YBTLB (95% CI: −2.07 to 10.95), HT (95% CI: −1.59 to 0.43), SSS20 (95% CI: −0.31 to 0.35), and CD10X5 (95% CI: −3.13 to −0.57). Significant improvements with meaningful precision were confirmed for LRSP, LASP, SRS, 20LSPT, and CD10X5.

## 4. Discussion

The findings of this study reinforce previous research, demonstrating that acrobatic and jumping programs significantly enhance motor abilities in adolescents. These improvements reflect the positive effects of specialized, movement-rich interventions that engage children in diverse, complex, and developmentally appropriate physical tasks. This aligns with outcomes from Hes and Asienkiewicz [30], who observed notable improvements in flexibility, upper body strength, and agility through targeted gymnastic activity in mixed-sex elementary school groups. Similarly, Petrušič and Novak [31] showed that even a modest extracurricular program (two additional sessions per week) led to significant gains in coordination and agility, highlighting how focused physical literacy interventions can yield measurable benefits in short timeframes. The findings of the recent meta-analysis by Taneja et al. [32], which demonstrated significant improvements in muscular strength, agility, flexibility, and balance in school-aged and adolescent gymnasts, strengthen the theoretical foundation of our study and emphasize the relevance and effectiveness of acrobatics-based interventions in adolescent PE.

In our study, flexibility improved significantly in the experimental group, corroborating prior findings on the efficacy of acrobatic routines for enhancing joint range of motion and muscular elasticity [4,8]. Specifically, LRSP scores increased from pre- to post-test (EG: 94.92 ± 11.39; CG: 83.38 ± 8.60, *p* < 0.001), with ANCOVA confirming a large effect (*p* < 0.01, η^2^ = 0.68; 95% CI: 5.95 to 17.13). LASP improvements were even more pronounced (EG: 132.29 ± 17.70 vs. CG: 104.96 ± 16.25, *p* < 0.001; ANCOVA means: EG = 133.67, CG = 103.58, η^2^ = 0.83; 95% CI: 17.91 to 36.75), while the control group experienced a minor decline (−5%, ES = −0.334). These results underscore the specificity and effectiveness of the experimental intervention.

Coordination gains were substantial, especially for the 20LSPT test. Although the experimental group had lower baseline scores (*p* = 0.021), post-test values significantly outperformed the control group (EG: 34.67 ± 6.31; CG: 45.16 ± 5.64, *p* < 0.001), and ANCOVA affirmed a strong group effect (*p* < 0.01, η^2^ = 0.82; 95% CI: −13.81 to −7.17). This result aligns with the literature linking movement complexity with improved coordination [6,8,33], a hallmark of acrobatic training. BOC results showed a non-significant raw group difference (*p* = 0.290), but ANCOVA revealed a small yet significant effect (*p* = 0.02, η^2^ = 0.12; 95% CI: −1.88 to 6.66), indicating modest benefits for fine motor control.

In contrast, SRS performance declined significantly in the experimental group, contrary to expectations. Post-test values dropped (EG: 70.96 ± 10.67 vs. CG: 82.13 ± 10.28, *p* < 0.001), and ANCOVA confirmed a large, statistically significant negative effect (EG = 69.45, CG = 83.63, *p* < 0.01, η^2^ = 0.71; 95% CI: −16.98 to −5.36). This may reflect motor interference or the absence of shoulder-specific mobility exercises within the program. It is also possible that accumulated fatigue or muscle tightness due to training load impaired performance. Future iterations should consider integrating dynamic shoulder mobility drills to counterbalance these effects.

Based on the results presented in Table 2 and Table 3, the CG showed maintenance of shoulder mobility (SRS), as indicated by nearly identical pre- and post-test mean values (82.38 cm vs. 82.13 cm) and no statistically significant change (*p* = 0.000 for between-group difference, but stable within-group values). This stability suggests that the standard PE curriculum likely included general activities, such as throwing, shooting, or spiking, that helped preserve shoulder range of motion. Consequently, while the EG showed a substantial improvement in SRS, the between-group difference may be partially attributed to the fact that the CG did not experience a decline in mobility, thereby narrowing the contrast in relative gains.

Balance, a key element of postural control and injury prevention, improved notably. For YBTUB, although pre-test values favored the CG (*p* = 0.007), post-test scores equalized (*p* = 0.742), and ANCOVA revealed a moderate group effect (EG = 107.58, CG = 99.56, *p* < 0.01, η^2^ = 0.41; 95% CI: −4.63 to 6.55). YBTLB also improved (ANCOVA: EG = 90.51, CG = 82.48, *p* < 0.01, η^2^ = 0.28; 95% CI: −2.07 to 10.95), with a 10% gain and ES = 0.629. These findings correspond with prior research emphasizing the role of acrobatic activities in developing balance through dynamic and single-leg control [4,8].

Speed improvements were more nuanced. The HT test showed no significant raw difference (*p* = 0.278), but ANCOVA revealed a small group effect (EG = 12.32, CG = 12.85, *p* = 0.02, η^2^ = 0.13; 95% CI: −1.59 to 0.43), suggesting some benefit in acceleration ability. However, SSS20 results remained unchanged (ANCOVA: *p* = 0.30, η^2^ = 0.03; 95% CI: −0.31 to 0.35), aligning with the notion that sprinting improvements require more specific drill exposure [6,33].

Agility, assessed through the CD10X5 test, improved significantly in the EG (post-test: EG = 20.43 ± 2.09, CG = 22.28 ± 2.52, *p* = 0.008), with ANCOVA confirming significance (EG = 20.67, CG = 22.04, *p* < 0.01, η^2^ = 0.21; 95% CI: −3.13 to −0.57). This aligns with previous findings on agility gains from multi-directional, reactive movement training [4,6,8].

Despite the EG’s broad improvements, the decline in SRS warrants attention. With a 95% confidence interval spanning −16.98 to −5.36, this finding indicates a true and meaningful decrease, potentially linked to intervention specificity or recovery inadequacies. Future studies should modify programming to protect or enhance fine motor mobility alongside gross motor gains.

Furthermore, the control group showed unexpected gains in HT and BOC. For HT, while not statistically significant (*p* = 0.278), the 95% CI (−1.59 to 0.43) suggests that practice effects or natural neuromuscular maturation may explain this result. Similarly, BOC improvements in CG (95% CI: −1.88 to 6.66) could reflect test familiarity or developmental trajectories. These findings stress the importance of including appropriate control groups and accounting for confounding factors to ensure valid longitudinal analysis.

Overall, the EG demonstrated significantly greater improvements in key motor abilities, especially flexibility (LRSP, LASP), coordination (20LSPT), balance (YBTUB, YBTLB), and agility (CD10X5), which are central to acrobatic training. These findings support the growing consensus that well-structured, developmentally appropriate PE programs can markedly boost motor competence in youth [4,6,8,33].

Limitations of this study include the exclusive use of male participants, which limits the generalizability of the findings. Future research should include mixed-gender cohorts to explore potential differential effects. Additionally, extracurricular PA was not monitored, which may have influenced the outcomes. Future studies should track external physical activity to better isolate the effects of the intervention. The absence of assessor blinding during pre- and post-testing introduces potential bias, despite the use of standardized protocols. Methodological heterogeneity between the EG and CG—due to differing PE content—also presents a limitation. Future research should aim to ensure comparable training loads and motor content across groups. Furthermore, the lack of follow-up testing restricts insight into the long-term sustainability of the program’s effects. Future studies should incorporate long-term follow-up assessments and include broader, more diverse school contexts to enhance the external validity of the findings. Improved control over measurement procedures and participant adherence will also strengthen internal validity and provide more robust evidence for the curricular integration of such programs.

## 5. Conclusions

This experimental study confirmed that an acrobatics-based PE program conducted two times per week significantly enhanced the development of key motor abilities—flexibility, coordination, balance, and agility—in adolescents compared to a standard PE curriculum. The results also suggest limited but task-specific improvements in speed, aligning with the hypothesis that effects on this variable would vary depending on the nature of the motor task.

PE teachers can adopt a progressive acrobatics module into the official school setting, demonstrating that such specialized interventions adapted to age, sex and skill levels can be feasibly implemented and produce measurable, superior outcomes in motor development. To ensure effective implementation, teacher training, appropriate equipment, and safety protocols should be emphasized.

## Figures and Tables

**Figure 1 life-15-00778-f001:**
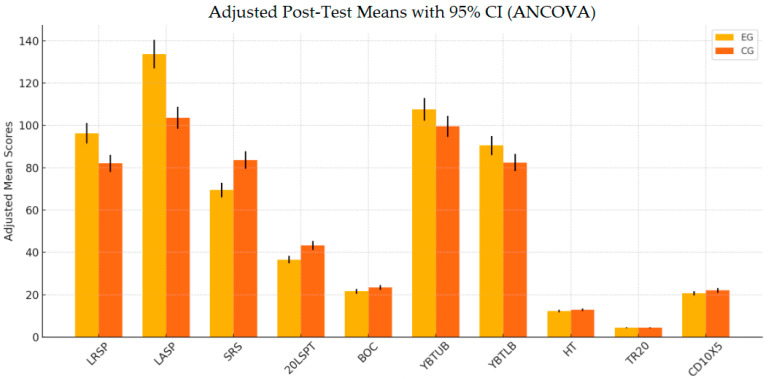
Adjusted post-test means for both the EG and CG derived from ANCOVA. Legend: The bar chart shows the adjusted post-test means with approximate 95% confidence intervals for both the experimental group (EG) and control group (CG).

**Table 1 life-15-00778-t001:** Sample characteristics at initial and final measurement.

	EG (*n* = 25)		CG (*n* = 25)	
	Initial	Final	Initial	Final
Age [years]	14.00 ± 0.50	14.33 ± 0.50	14.00 ± 0.50	14.33 ± 0.50
Body Mass [kg]	59.87 ± 20.00	63.99 ± 22.00	58.88 ± 20.00	64.08 ± 17.00
Body Height [cm]	162.53 ± 10.00	167.69 ± 8.00	160.26 ± 10.00	164.53 ± 7.00
BMI [kg/m^2^]	22.65 ± 6.43	22.75 ± 6.96	22.95 ± 6.38	23.67 ± 5.55

Legend: Values are presented as mean ± standard deviation; EG–experimental group; CG–control group.

**Table 2 life-15-00778-t002:** Descriptive statistics and comparison of initial and final measurements between the experimental and control groups.

Variable	Pre-Test						Post-Test					
	EG		CG				EG		CG			
	Mean	SD	Mean	SD	t	Sig.	Mean	SD	Mean	SD	t	Sig.
LRSP (deg)	77.38	10.19	80.50	9.82	−1.081	0.285	94.92	11.39	83.38	8.60	3.961	0.000
LASP (deg)	107.33	15.40	109.92	13.30	−0.622	0.537	132.29	17.70	104.96	16.25	5.571	0.000
SRS (cm)	85.88	12.74	82.38	10.63	1.033	0.306	70.96	10.67	82.13	10.28	−3.692	0.000
20LSPT (s)	45.75	8.66	50.92	6.15	−2.35	0.021	34.67	6.31	45.16	5.64	−6.076	0.000
BOC (s)	30.18	9.38	25.37	9.43	1.770	0.083	23.69	7.45	21.30	7.96	1.071	0.290
YBTUB (%)	95.04	11.95	103.59	8.70	−2.832	0.007	104.05	11.47	103.09	8.48	0.331	0.742
YBTLB (%)	80.59	13.29	82.54	12.68	−0.519	0.606	88.71	12.53	84.27	10.90	1.311	0.196
HT (s)	15.00	2.35	15.06	2.63	−0.082	0.935	12.30	1.88	12.88	1.77	−1.098	0.278
SSS20 (s)	4.61	0.70	4.64	0.71	−0.123	0.903	4.43	0.57	4.41	0.62	0.126	0.901
CD10X5 (s)	23.22	2.22	23.88	2.15	−1.055	0.297	20.43	2.09	22.28	2.52	−2.765	0.008

Legend: EG–experimental group; CG–control group; Mean–average value; SD–standard deviation; Sig.–statistical significance (*t*-test); *p* < 0.05.

**Table 3 life-15-00778-t003:** Univariate analysis of covariance of experimental and control groups.

Variable	Adj. Mean EG	Adj. Mean CG	Adj. Mean Diff. (E-K)	F	Sig.	η^2^
LRSP (deg)	96.25	82.04	14.21	90.38	0.00	0.68
LASP (deg)	133.67	103.58	30.08	205.54	0.00	0.83
SRS (cm)	69.45	83.63	−14.19	106.37	0.00	0.71
20LSPT (s)	36.60	43.23	−6.63	187.84	0.00	0.82
BOC (s)	21.57	23.42	−1.84	5.69	0.02	0.12
YBTUB (%)	107.58	99.56	8.01	31.16	0.00	0.41
YBTLB (%)	90.51	82.48	8.03	16.97	0.00	0.28
HT (s)	12.32	12.85	−0.52	6.24	0.02	0.13
SSS20 (s)	4.44	4.40	0.04	1.11	0.30	0.03
CD10X5 (s)	20.67	22.04	−1.37	10.77	0.00	0.21

Legend: Adj. Mean—adjusted arithmetic mean (EG—experimental group, CG—control group); Adj. Mean diff.—differences between adjusted arithmetic means; F—F-test; *p*–significance level (*p* < 0.05 or *p* < 0.01 indicates statistical significance); partial eta squared effect size: η^2^ = 0.01—small effect, η^2^ > 0.06—moderate effect, η^2^ > 0.14—large effect.

## Data Availability

The data provided in this study can be obtained upon request from the corresponding author.

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
