# Peer review of "The Impact of an Acrobatics-Based Curriculum on Motor Fitness in Adolescents"

_life, 2025, doi:10.3390/life15050778_

Round 1
Reviewer 1 Report
Comments and Suggestions for Authors
Dear Authors
If is possible in the introduction add more information regarding School-based PE programs to increase fitness, social and mental capacity. Also, about the contributions of the acrobatics programs in muscular and aerobic capacity.
Methods
Did you control the physical activity in control group or experimental? in clubs, fitness centers?
Discussion
About the significant results acrobatics program can you add recent studies that show us the effectiveness of jump, for example, comparatively to the standard PE curriculum?
Author Response
Dear Reviewer,
Thank you very much for taking the time to review this manuscript and for contributing to its substantial improvement.
Please find the detailed responses attached and the corresponding revisions/corrections highlighted in the resubmitted file.
Kind regards,
The authors

Reviewer 2 Report
Comments and Suggestions for Authors
See attachement.

Author Response

(The authors gave the same response as above.)

Reviewer 3 Report
Comments and Suggestions for Authors
See attachment

Author Response

(The authors gave the same response as above.)

Reviewer 4 Report
Comments and Suggestions for Authors
The presented article is devoted to topical issue of optimizing physical education at school. The selection of effective forms of physical exercise classes, which may be interesting for children, has a positive effect on health and physical fitness of schoolchildren. In addition, these classes can become the basis for further independent classes.
The manuscript is presented in a well-structured form. The description of research methods allows you to reproduce its results.
The data are displayed in an accessible form, it is structured, that allows you to clearly reproduce obtained results.
Results correspond to conclusions and have significant theoretical and practical significance.
Here are some discussion questions that can contribute to improving the article:
- The author draws attention to differences in the level of physical activity between adolescents of different sexes, namely, stating that teenage girls are less active (line 72). It requires an explanation why boys participated in the study (line 119), although these are girls that need active involvement in classes.
- Arrangement of classes needs explanation. Was it 1 group (25 students) who were engaged simultaneously, or were they divided into 2 groups? In case of one group, how many teachers worked with this group?
- Conceptual apparatus of the study needs clarification, namely, “balance”, “coordination” needs unification. According to the literature, the ability to maintain balance “balance” is one of the forms of coordination abilities. Therefore, it is necessary to understand which of the other forms of coordination abilities the author meant by the term “coordination”.
- It is necessary to clarify how many times a week the classes were held – 2 times (line 122) or three times a week (line 90).
- There is a question regarding appropriateness of mentioning positive impact of classes on health and body composition, if these data are not presented in the results of the study.
- It is advisable to expand the list of used reference sources, adding publications over the past 5 years.
- In the tables, units of measurement of indicators should be provided, for example, in table 1 "Age", tables 3, 4 for all indicators. Specify the value of note in table 3 p < 0.05, < 0.01., as shown in the table.
Author Response

(The authors gave the same response as above.)

Reviewer 5 Report
Comments and Suggestions for Authors
The authors address a topic of high educational and public health interest: the effect of an acrobatics programme on components of motor fitness. They justify the intervention based on an up-to-date review of the benefits of physical exercise in adolescents, incorporating key studies. The authors formulate a clear hypothesis, anticipating improvements in flexibility, coordination, balance, and agility, with variable effects on speed.
Although a previous publication is mentioned, it is not clearly stated what the actual novelty is in relation to the earlier study. This may raise doubts about the originality of the manuscript. There is a lack of specific references to recent meta-analyses or systematic reviews that could serve as contrast and theoretical reinforcement (especially in relation to acrobatics-based programmes). Therefore, it is recommended to clearly state the novel contribution of this study in relation to the previous one and to strengthen the theoretical framework with recent systematic reviews.
With regard to the methodology section, the authors provide a detailed description of the sample, inclusion/exclusion criteria and the randomisation procedure. The quasi-experimental design is well structured, with 32 sessions, controlled environment, and well-defined phases in each session. They also make use of a variety of well-referenced assessment instruments for flexibility, coordination, balance, speed, and agility.
However, the authors present a study with a limited number of participants (n=50) for the desirable statistical power in a multivariate ANCOVA analysis. No power calculation or justification of sample size is reported. It is also not specified whether there was blinding of assessors, which is relevant to reduce bias in functional tests. Moreover, the intervention is compared to conventional physical education classes (handball and volleyball), which introduces certain methodological heterogeneity in terms of external variable control (e.g., actual physical load, motivation, motor content). It is therefore recommended that the authors justify the sample size, report the achieved statistical power and discuss the limitations of the control group in terms of comparability.
Regarding the results section, the authors present them in an organised way and support them with tables (Tables 3 and 4) and explanatory graphs (Figure 1) that facilitate understanding. Appropriate statistical analyses are used: t-tests, ANCOVA, effect size (Cohen and η²), with interpretation according to standard criteria, and significant improvements are observed in most variables in the experimental group (flexibility, coordination, balance, agility).
However, there are certain aspects in this section that should be improved or elaborated upon. Firstly, confidence intervals for the effects are not reported, which limits interpretative precision. The SRS variable (shoulder rotation) shows a negative result in the experimental group without detailed explanation. In some cases, the control group also improves significantly (HT, BOC), but these findings are not sufficiently discussed or explained. As a consequence of these aspects, the authors should include confidence intervals, analyse unexpected results (such as the decline in SRS) and explain improvements in the control group to strengthen internal validity.
In terms of the discussion presented by the authors, it is well structured and links the findings with previous research. The effectiveness of the programme in improving flexibility and agility is appropriately highlighted, with supporting references. In addition, the authors acknowledge limitations in the effect on speed, especially in the short sprint (SSS20).
However, the explanation for the decline in the SRS test in the experimental group is insufficient. Possible measurement errors or joint fatigue effects are also not discussed. The discussion is very positive, without addressing possible methodological biases, such as lack of follow-up, interindividual variability or lack of measurement of effective adherence. The authors also do not address specific pedagogical implications or how to integrate this type of programme into the official curriculum. Therefore, it would be advisable for the authors to discuss possible methodological biases, interpretation errors and to offer practical guidance for teachers.
Finally, the conclusions section clearly and coherently summarises the results and highlights the potential of acrobatics programmes to improve multiple motor skills. However, it makes strong claims (“highly effective”) without qualifying that effects vary between abilities and that the impact on speed was limited. For this reason, the authors should formulate more balanced conclusions, acknowledging both strengths and limitations, and include future improvement lines (long-term follow-up, inclusion of girls, generalisation to other school contexts).
Therefore, it is recommended that the following improvements be made before this work can be published:
- Justify the sample size and include power analysis.
- Better explain the performance decline in SRS and other atypical results.
- Discuss possible biases, practical implications and generalisation.
- Include confidence intervals and improve the pedagogical discussion.
- Moderate the final conclusions to better reflect the diversity of effects.
Author Response

(The authors gave the same response as above.)

Round 2
Reviewer 3 Report
Comments and Suggestions for Authors
Please find the attached document.

Author Response
Dear Reviewer,
Thank you once again for taking the time to review this manuscript and for contributing to its substantial improvement.
Please find the detailed responses attached and the corresponding revisions/corrections highlighted in the resubmitted file.
Kind regards,
The authors
